# Comparison of Kinematics and Electromyographic Activity in the Last Repetition during Different Repetition Maximums in the Bench Press Exercise

**DOI:** 10.3390/ijerph192114238

**Published:** 2022-10-31

**Authors:** Stian Larsen, Markus Haugen, Roland van den Tillaar

**Affiliations:** Department of Sport Sciences and Physical Education, Nord University, 8026 Levanger, Norway

**Keywords:** bench press, biomechanics, sticking region, EMG

## Abstract

The barbell bench press is often performed at different repetition maximums (RM). However, little is known about the last repetition of these repetition maximums in terms of movement kinematics and electromyographic activity in the bench press. This study compared kinematics and electromyographic activity during the last repetition of 1-RM, 3-RM, 6-RM, and 10-RM on the barbell bench press. Twelve healthy recreationally bench press-trained males (body mass: 84.3 ± 7.8 kg, age: 23.5 ± 2.6 years, height: 183.8 ± 4.2 cm) performed the bench press with a self-chosen grip width with four different repetition maximums. The participants bench pressed 96.5 ± 14.1, 88.5 ± 13.0, 81.5 ± 12.3, and 72.8 ± 10.5 kg with the 1-RM, 3-RM, 6-RM, and 10-RM. No differences were found between the bench press conditions in kinematic or electromyographic activity, except for the 10-RM, where a higher barbell velocity was observed at peak barbell deacceleration and first minimum barbell velocity (*p* ≤ 0.05) compared to the 1-RM and 3-RM. Overall, triceps medialis activity increased, whereas biceps brachii activity decreased from the pre-sticking to post-sticking region for all bench conditions (*p* ≤ 0.05). Since slower barbell velocity was observed in the sticking region for the 1-RM and 3-RM conditions compared to the 10-RM condition, we suggest training with these repetition maximums to learn how to grind through the sticking region due to the principle of specificity when the goal is to enhance maximal strength.

## 1. Introduction

The barbell bench press exercise is a popular exercise for enhancing the strength of the upper limbs. In the sport of powerlifting, the one-repetition maximum (1-RM) is evaluated in the squat, bench press, and deadlift, where the highest approved load lifted in each exercise is added to the total [1].

There are several ways to train for increasing strength and hypertrophy of the musculoskeletal system. The prevailing theory has suggested that enhancement of maximal strength is improved when training with heavy loads, [2], whereas hypertrophic response may be achieved at both high and low loads [3]. However, training with loads to concentric exhaustion can be performed with different numbers of repetition maximums (RM), meaning training to full exhaustion at for example 1-RM, 3-RM, 6-RM and 10-RM.

Moreover, Schoenfeld et al. [4] investigated muscular adaptations and strength in response to a volume-equated bodybuilding program (three sets of 10-RM) vs. a typical powerlifting program (seven sets of 3-RM) in 17 young well-trained men. The authors found that the 3-RM group increased their 1-RM bench press to a greater extent than the 10-RM group. This is logical because the ability to produce force is a result of a combination of muscle mass [5], neural factors [6], and the specificity of the load itself during resistance exercises [7]. However, the body of knowledge about how the kinematics is affected by the load when the set is taken to concentric failure is, to our understanding, not well studied or understood for the bench press exercise.

Failure may occur due to acute fatigue mechanisms such as excitation-contraction coupling failure. However, both joint and barbell kinematics may fluctuate as fatigue increases. This is already indicated by the finding that a sticking region only occurs at loads of more than 85% of 1-RM [8], but also in the last repetition during 6-RM [9]. However, van den Tillaar et al. [9] did not compare the kinematics and electromyographic (EMG) activity directly between 1-RM and 6-RM.

Even though EMG activity may be similar according to the Henneman size principle [10] because this is a proxy for muscle excitation and muscle excitation would be expected to be similar when the set is taken close to concentric failure, differences in movement kinematics may occur. This is because fatigue may arise non-locally, meaning that remote non-exercised muscle(s) and the endocrine or cardiovascular systems may also be fatigued [11]. However, the non-local muscle fatigue literature is conflicting, and four different pathways (neurological, biomedical, psychological and biomechanical) may affect non-local muscle fatigue [12]. For example, a longer set duration with lower loads compared to a shorter set duration with higher loads may play a role in the biomechanical pathway, which may lead to different intraset technique alterations and observed movement kinematics.

Moreover, it is likely that higher barbell loads in the bench press will result in a larger moment of inertia compared to lower barbell loads, assuming the radiuses of gyration are similar. It is speculated that this may lead to slower barbell velocities because a larger moment of inertia will increase the time to accelerate the barbell even if the set is taken to maximal effort. Potentially, this means that using low load repetition maximums, such as 10-RM (+), may not be specific from a biomechanical or kinematic standpoint.

Potentially slower barbell velocities for high-load sets could also result in lower barbell displacement and joint angles in and around the sticking region. Therefore, the purpose of this study was to investigate the effects of 1-RM, 3-RM, 6-RM, and 10-RM on kinematics and electromyographic activity in the barbell bench press during the last repetition. It was hypothesized that a sticking region would occur during all RMs, but that barbell velocity would be lower with the 1-RM and 3-RM compared to the 6-RM and 10-RM because of a larger inertia/moment of inertia. Also, it was hypothesized that no differences in electromyographic activity would occur because muscle excitation would be expected to be similar when the set is taken close to concentric failure, according to the Henneman size principle [10].

## 2. Materials and Methods

### 2.1. Experimental Approach to the Problem

A cross-sectional repeated measurement study with within-subject design was used to investigate differences in kinematics and electromyographic activity of the last repetition with 1-RM, 3-RM, 6-RM, and 10-RM in the bench press exercise. The study included three familiarization sessions and one test session, with a minimum of 72 h between each session. The dependent variables—barbell kinematics and joint kinematics—were collected as means during the events: lowest barbell height (v_0_), first peak barbell velocity (v_max1_), first peak barbell deacceleration (d_max1_), first minimum barbell velocity (v_min_), and second peak barbell velocity (v_max2_). Electromyographic activities were collected as means during the pre-sticking (v_0_–v_max1_), sticking (v_max1_–v_min_) and post-sticking regions (v_min_–v_max2_).

### 2.2. Participants

Twelve recreationally bench-press-trained men volunteered for the study (body mass: 84.3 ± 7.8 kg, age: 23.5 ± 2.6 years, height: 183.8 ± 4.2 cm). The participants had on average 5.7 ± 2.1 years of strength training experience (at least twice a week strength training). Inclusion criteria were: (1) previously self-reported 1-RM at 1.0 times body mass in the last 6 months; (2) injury-free, so maximum performance was not reduced; and (3) age between 18 and 40 years. Furthermore, participants were not allowed to consume alcohol or train the upper extremities for 48 h before testing, and no caffeine was allowed during the test session. The participants were informed orally and in written format about possible risks. Written consent for each participant was obtained before the first familiarization session. The study was performed according to institutional ethical requirements, and approval for data security and handling was obtained from the Norwegian Center for Research Data (project nr: 701688) and in accordance with the latest revision of the Declaration of Helsinki.

### 2.3. Procedures

In familiarization session one, height, body mass, age, training experience, and grip width were measured. The participants could self-select grip width, but they had to adhere to it during all sessions and for all lifts. White tape was placed on the bar to indicate the placement of the index finger for all sessions. The grip width was measured as the distance between the index fingers. Lifting equipment such as lifting belts, wrist support, elbow support and chalk were not allowed. Participants tested 1-RM, 3-RM, 6-RM, and 10-RM at all three familiarization sessions and the test session, but in a randomized order (order determined using random.org, accessed on 1 October 2022). The warm-up procedure followed the protocol of Gomo et al. [13] during all four sessions. The barbell (Ohio Power Bar; Rogue Fitness, Columbus, OH, USA) had to be unracked without assistance. For the repetitions to be counted as approved, the participant needed to lower the barbell without bounds and fully extend the arms and elbows. Barbell velocity was measured in the 1-RM, 3-RM, 6-RM, and 10-RM lifts during the familiarization sessions with a linear encoder (ET-Enc-02, Ergotest Technology AS, Langesund, Norway) to determine whether the actual RMs (similar velocity at last repetition) were achieved in the test session. Participants were allowed to rest for 4 min between maximal attempts. In the test session, the participants performed the same warm-up protocol as Gomo et al. [13] on the familiarization day. When a participant successfully lifted a previous RM, the load was increased by 0.25–5 kg until the participant reached concentric failure.

A linear encoder (ET-Enc-02, Ergotest Technology AS, Langesund, Norway) measuring with a resolution of 0.075 mm and counting the pulses with 10 ms intervals was used to determine maximal barbell velocity and lifting time of the barbell. The vertical displacement was measured as distance in meters from v_0_ to full extension. The shoulder and elbow joint angles, and barbell displacement and velocity were recorded during the ascent of the last repetition v_0_, v_max1_, d_max1_, v_min_, and v_max2_. The linear encoder was synchronized with the EMG recordings using a Musclelab 6000 system and analyzed using Musclelab version 10.200.90.5095 (Ergotest Technology AS, Langesund, Norway).

EMG activity was recorded using Musclelab 6000 (Ergotest Technology AS, Langesund, Norway). After the skin was shaved and rubbed, a small amount of conductive gel was applied to the electrodes. Electrodes of 11 mm contact diameter and 2 cm center-to-center distance (Zynex Neurodiagnostics, Lone Tree, CO, USA) were attached to seven muscles (the clavicular and sternocostal parts of the pectoralis major, anterior and lateral head of the deltoid, lateral and medial head of the triceps, biceps brachii and latissimus dorsi) in the assumed orientation of the underlying muscle fiber, according to the recommendations of SENIAM [14]. The raw EMG signals were converted to root of means squared (RMS) by a hardware circuit network (frequency response 20–500 kHz, averaging constant 100 ms, total error ± 0.5%). The data were sampled at a rate of 1000 Hz. The data were analyzed using commercial software (Musclelab V8.13, Ergotest Technology AS, Langesund, Norway). The mean RMS was calculated for the pre-sticking, sticking and post-sticking regions.

A three-dimensional (3D) motion capture system (Qualisys, Gothenburg, Sweden) with eight cameras sampling at a frequency of 500 Hz was used to track reflective markers, creating 3D positional measurements. Markers were placed on both sides of the body. A total of 14 markers were placed on the distal part of the radius and ulna, lateral and medial epicondyle of the humerus, the acromion, and sternum. Markers were also placed at both ends of the barbell, including two markers right inside the plate loading. Motion capture files were exported as C3D files to Visual 3D (C-Motion, Inc., Washington, DC, USA) for modelling and kinematic analyses. Variables were lowpass filtered with a Butterworth filter with a cut-off frequency of 6.0 Hz. Variables analyzed in Visual 3D were joint angles and joint angular velocities for the elbow and shoulder joints.

### 2.4. Statistical Analyses

To compare barbell and joint kinematics, a four (condition: 10-RM, 6-RM, 3-RM, and 1-RM) by five (event: v_0_, v_max1,_ d_max1_, v_min_ and v_max2_) analysis of variance (ANOVA) with repeated measures was used. To compare electromyographic activity, a four (condition: 10-RM, 6-RM, 3-RM, and 1-RM) by three (region: pre-sticking, sticking, and post-sticking) ANOVA for each of the muscles was applied. A Bonferroni post hoc test was used to identify potential differences in kinematics or electromyographic activity. If the sphericity assumption was violated, *p*-values of the Greenhouse-Geisser adjustment were reported. Every result is presented as mean ± standard deviation (SD). The alpha level was set at *p* < 0.05. Effect size was evaluated as η_p_^2^ (eta partial squared), where 0.01 < η_p_^2^ < 0.06 constitutes a small effect, 0.06 < η_p_^2^ < 0.14 constitutes a medium effect, and η_p_^2^ > 0.14 constitutes a large effect [15]. Statistical analyses were performed in IBM SPSS Statistics 27.0 (IBM, Armonk, NY, USA).

## 3. Results

The participants lifted 96.5 ± 14.1, 88.5 ± 13.0, 81.5 ± 12.3 and 72.8 ± 10.5 kg in the 1-RM, 3-RM, 6-RM, and 10-RM bench press. A significant difference between bench press conditions was found for the events v_max1_ and d_max1_ (F ≥ 3.85, *p* ≤ 0.04, η_p_^2^ ≥ 0.26), but not for v_min_ and v_max2_ (F ≤ 1.25, *p* ≥ 0.31, η_p_^2^ ≤ 0.1, Table 1). Post hoc tests revealed that velocity was higher at v_max1_ and d_max1_ for the 10-RM compared with the 1-RM and 3-RM (*p* < 0.027). No differences were found between the conditions for lifting time (F ≤ 1.35, *p* ≥ 0.27, η_p_^2^ ≤ 0.11, Table 1).

No differences in barbell height or horizontal displacement were found between bench press conditions for any of the events (F ≤ 0.66, *p* ≥ 0.5, η_p_^2^ ≤ 0.056). For horizontal displacement, a significant effect was found for region (F = 95.74, *p* < 0.001, η_p_^2^ = 0.9), where post hoc tests revealed that the horizontal displacement from the shoulder origin decreased for each event (*p* < 0.039) (Figure 1 and Table 1).

No significant differences were found between the conditions for shoulder flexion, shoulder abduction or elbow extension angles in any of the events (F ≤ 2.8, *p* ≥ 0.058, η_p_^2^ ≤ 0.24, Figure 2).

The elbow and shoulder reached peak angular velocity elbow extension and shoulder flexion at approximately the first and second peak barbell velocity (Figure 3), and their velocity decreased between these points, with their minimal angular velocities concomitant with the event v_min_. Moreover, peak shoulder abduction velocity occurred only once. No significant effect of bench press condition was found for elbow extension velocity or shoulder abduction velocity (F ≤ 0.7, *p* ≥ 0.54, η_p_^2^ ≤ 0.08). However, a significant effect was found for shoulder flexion velocity (F = 5.07, *p* = 0.032, η_p_^2^ = 0.39). Post hoc tests revealed greater first shoulder peak velocity when benching with a 10-RM compared to the other bench press conditions (Figure 3).

For electromyographic activity, no significant differences were found between the bench press conditions or for region (F ≤ 2.4, *p* ≥ 0.096, η_p_^2^ ≤ 0.25, Figure 4), except for regions for the triceps medialis and biceps brachii (F ≥ 3.6, *p* ≤ 0.05, η_p_^2^ ≥ 0.34). Here, post hoc tests revealed that the triceps medialis increased electromyographic activity from the pre-sticking to post-sticking region, whereas the opposite occurred for the biceps brachii (Figure 4).

## 4. Discussion

The aim of this study was to investigate the effects of 1-RM, 3-RM, 6-RM, and 10-RM upon kinematics and electromyographic activity during the last repetition in the barbell bench press. The main findings were that greater barbell velocity was observed at v_min_ and d_max1_ for the 10-RM compared to the 1-RM and 3-RM, which partly confirms our hypothesis. Also, the 10-RM had a greater first peak in shoulder flexion angular velocity compared to the other bench press conditions. However, different RMs did not influence other kinematic measurements such as horizontal or vertical barbell displacement and joint angles. Moreover, no differences were found in electromyographic activity between bench press conditions, which is in line with our hypothesis. Also, the biceps brachii showed decreased electromyographic activity, whereas the triceps medialis showed increased electromyographic activity, from the pre-sticking to the post-sticking region.

A clear sticking region was observed for all bench conditions, which was in accordance with previous studies investigating biomechanics around the sticking region in the bench press exercise [9,16,17,18]. The sticking region started at approximately 0.033 m and ended at 0.290 m barbell height, with no differences between the bench conditions. Furthermore, as hypothesized, no differences were found in electromyographic activity for any of the measured muscles (Figure 4). This is in line with the Henneman size principle [10], and is logical because all muscles were taken to maximal concentric effort. However, greater barbell velocity was observed at v_max1_ and d_max1_ for the 10-RM compared to the others (Table 1), which occurred due to a greater first peak in shoulder flexion angular velocity for the 10-RM (Figure 3). It is speculated that greater barbell velocity for the 10-RM was observed due to the higher loads at 1-RM and 3-RM, and therefore, greater moment of inertia, because the distance from shoulder origin + and joint angles were similar (Figure 1 and Figure 2), meaning that moment arms were similar between the conditions. A larger moment inertia will increase the time to accelerate the barbell and may have been responsible for the slower barbell velocities observed for the low load conditions in our study. However, we cannot conclude that moment of inertia was similar between the conditions because the bench press has a horizontal force component, affecting the horizontal barbell moment arms [1,19]. Interestingly, higher barbell velocity for the 10-RM condition did not influence the rest of the observed barbell kinematics in our study, which means that bench pressing with a broad RM may result in similar movement kinematics.

The horizontal displacement from the shoulder origin decreased through the concentric for all bench conditions (Table 1 and Figure 1). This was achieved with an increased shoulder abduction angle (Figure 2), which is consistent with previous research [1,13]. Abducting the shoulder may reduce the external extension or horizontal abduction moment of the arm from the barbell to the shoulder origin, resulting in an easier lockout in the post-sticking region, as demonstrated in Figure 1. Interestingly, the first peak in shoulder flexion velocity ranged from approximately 45–60 °/s, whereas the first peak in elbow extension angular velocity ranged from 35–40 °/s, meaning that the bench press starts with larger shoulder flexion velocity than elbow extension velocity (Figure 3). Thereafter, almost midway between d_max1_ and v_min_, peak shoulder abduction velocity was observed. It is speculated that the participants self-organized to decrease the external shoulder moment of the arm, potentially making the lift easier on the shoulder flexor and horizontal abduction muscles. However, our EMG data did not confirm this speculation, which may be due to a more advantageous length-tension relationship for the sarcomeres for the pectoralis muscles [20]. Both the elbow extensors and shoulder flexors reached a minimum velocity, which was concomitant with v_min_. However, the elbow extensors reached a mean minimum angular velocity of approximately 0–2 °/s, whereas approximately 3–10 °/s minimum angular shoulder flexion velocity was observed for the shoulder flexors. Then, both the elbow extensors and shoulder flexors reached a second peak angular velocity, which was concomitant with v_max2_. An interesting finding was that the elbow extensors reached a larger second peak velocity (84–97 °/s) than the shoulder flexors (57–66 °/s).

The second peak velocity is consistent with our EMG data, showing an increase in triceps medialis activity from the pre-sticking to post-sticking region. This is in line with previous research showing increased electromyographic activity from the pre-sticking to the post-sticking region [16]. We speculate that the increase in electromyographic activity is due to the lateral force component in the barbell bench press. Even though the participants could self-select grip width, most of the participants benched with a wide grip width according to the standards of Larsen, et al. [1]. These investigators found that benching with a wide grip width resulted in an increase in lateral forces exerted against the bar from the pre-sticking to the post-sticking region and, therefore, greater triceps medialis electromyographic activity. Moreover, the opposite occurred for the biceps brachii muscle, where electromyographic activity decreased from the pre-sticking to post-sticking region. The increase in lateral forces during the post-sticking region increases triceps medialis activity, potentially leading to a reciprocal inhibition of the biceps brachii [21].

The present study has some limitations. Firstly, we only analyzed the last repetition, and not the whole process in terms of kinematics and EMG for all repetitions. Since training is often not taken to maximal effort, analyzing the full development may provide more detailed information about the movement strategies before maximal effort is reached. Also, we used recreationally trained lifters and not experienced powerlifters; studies should be conducted with stronger lifters to confirm our findings in these cohorts.

## 5. Conclusions

Since similar joint kinematics and electromyographic activity was observed between the bench press conditions, it was concluded that, from an acute functional anatomical standpoint, a wide range of RMs can be used when the goal is to increase maximal strength in the bench press exercise. However, for bench press athletes and powerlifters we recommend bench pressing with high load conditions such as 1-RM and 3-RM, to learn how to grind through the sticking region with slower barbell velocities.

## Figures and Tables

**Figure 1 ijerph-19-14238-f001:**
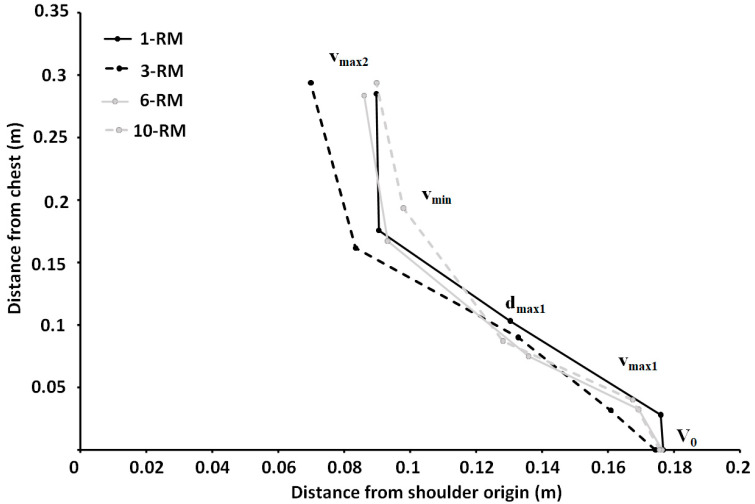
Mean distance in meters from the chest is shown on the y-axis and the distance from the shoulder origin on the x-axis for the 1-RM, 3-RM, 6-RM, and 10-RM in the bench press for the events v_0_, v_max1_, d_max1_, v_min_ and v_max2_.

**Figure 2 ijerph-19-14238-f002:**
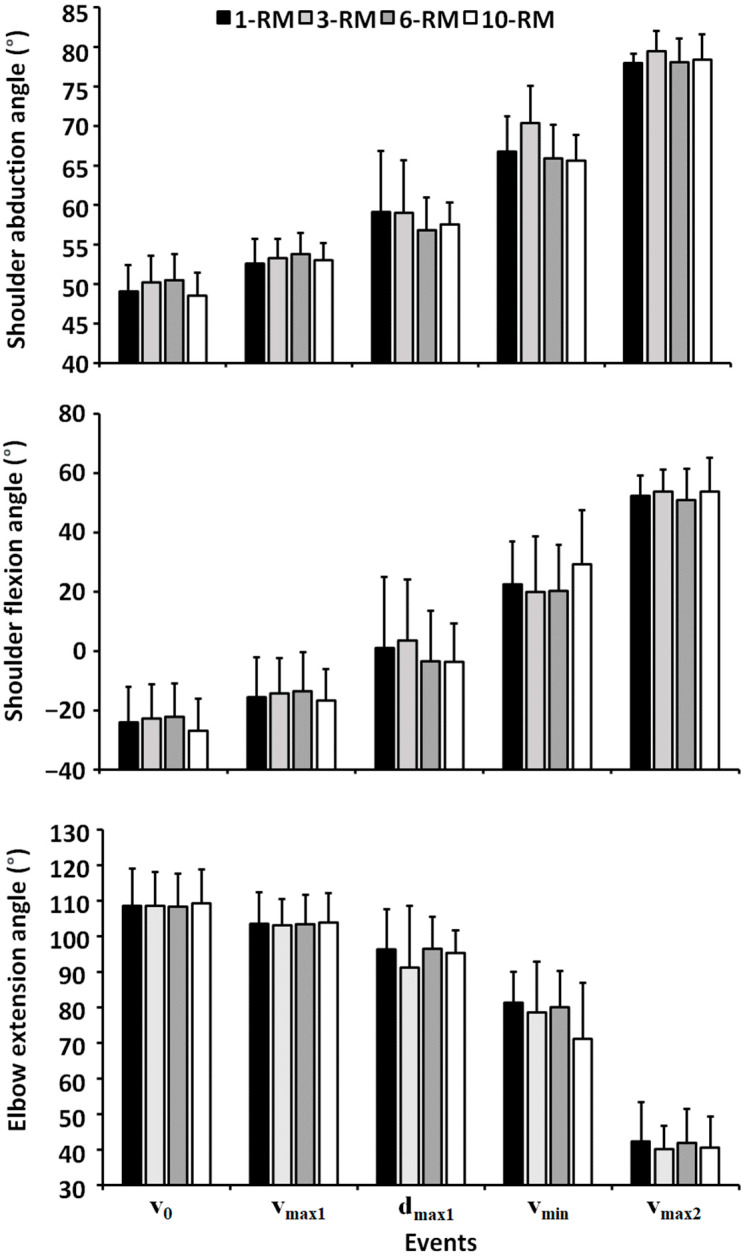
Mean ± SD shoulder abduction, shoulder flexion and elbow extension angle for the 1-RM, 3-RM, 6-RM, and 10-RM in the bench press for the events v_0_, v_max1_, d_max1_, v_min_ and v_max2_.

**Figure 3 ijerph-19-14238-f003:**
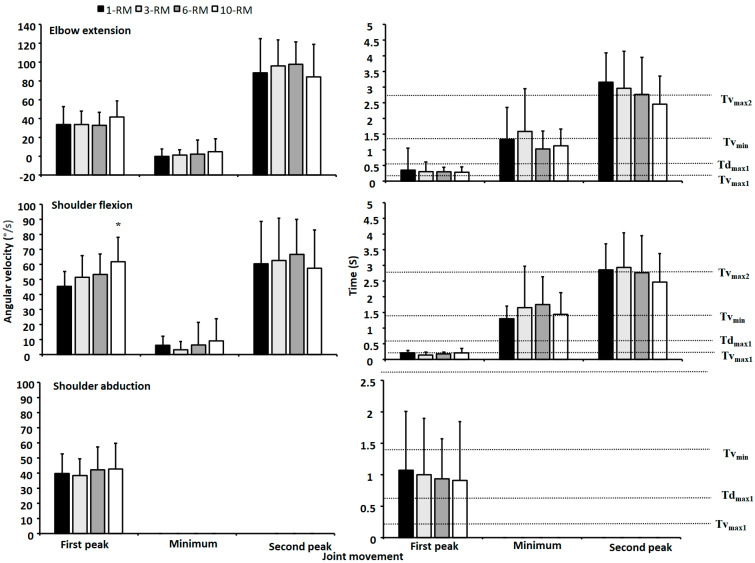
Peak and minimal mean ± SD joint movements for elbow extension, shoulder flexion and shoulder abduction velocity with their timings for the 1-RM, 3-RM, 6-RM, and 10-RM in the bench press exercise. Also shown is the mean timing of the events v_max1_, d_max1_, v_min_ and v_max2_ for all bench press conditions relative to the timing of the joint movements. * Indicates a significant difference in barbell velocity between the 10-RM compared with the 1-RM and 3-RM at this event in the bench press exercise (*p* < 0.05).

**Figure 4 ijerph-19-14238-f004:**
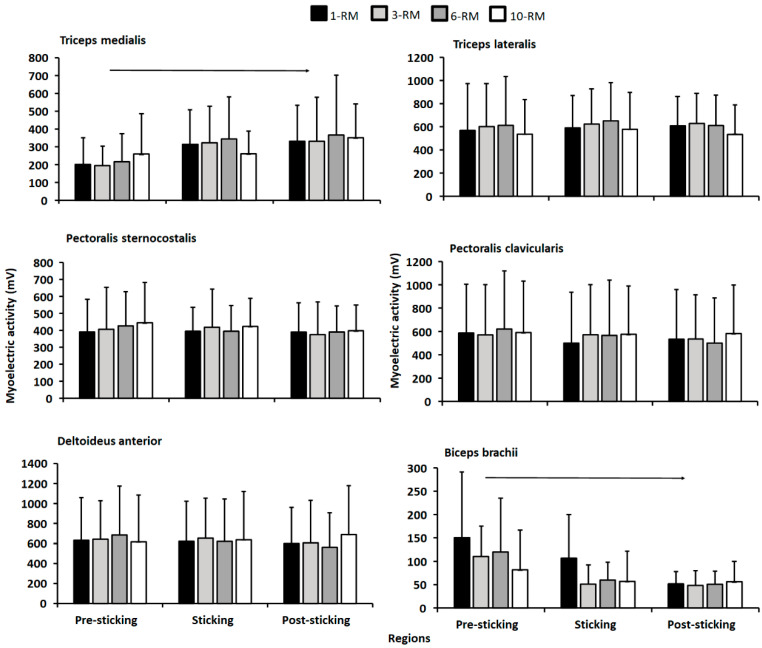
Mean ± SD electromyographic activity for the medial triceps, lateral triceps, sternocostal pectoralis major, clavicular pectoralis major, anterior deltoid and biceps brachii in the pre-sticking, sticking and post-sticking region for the 1-RM, 3-RM, 6-RM, and 10-RM bench press exercise. → Indicates a significant difference in electromyographic activity between the pre-sticking and the post-sticking region in the bench press exercise (*p* < 0.05).

**Table 1 ijerph-19-14238-t001:** Mean ± SD relative barbell height, barbell velocity, and time at different RMs at the events v_max1_, v_min_, and v_max2_ during bench press. ∗ *p* < 0.05.

Event	Condition	Barbell Height (m)	Horizontal Displacement (m)	Barbell Velocity (m/s)	Time (s)
v_max1_	1-RM	0.028 ± 0.01	0.176 ± 0.05	0.21 ± 0.06	0.21 ± 0.09
	3-RM	0.032 ± 0.015	0.160 ± 0.06	0.22 ± 0.08	0.19 ± 0.08
	6-RM	0.032 ± 0.013	0.169 ± 0.05	0.26 ± 0.06	0.2 ± 0.05
	10-RM	0.040 ± 0.015	0.167 ± 0.05	0.29 ± 0.09 *	0.22 ± 0.1
d_max1_	1-RM	0.103 ± 0.11	0.130 ± 0.05	0.15 ± 0.08	0.78 ± 0.78
	3-RM	0.090 ± 0.072	0.133 ± 0.05	0.16 ± 0.08	0.92 ± 1.36
	6-RM	0.075 ± 0.04	0.136 ± 0.05	0.20 ± 0.06	0.41 ± 0.27
	10-RM	0.087 ± 0.04	0.128 ± 0.06	0.21 ± 0.10 *	0.42 ± 0.20
v_min_	1-RM	0.0176 ± 0.07	0.090 ± 0.05	0.05 ± 0.03	1.48 ± 0.46
	3-RM	0.162 ± 0.083	0.084 ± 0.04	0.04 ± 0.04	1.45 ± 1.14
	6-RM	0.167 ± 0.059	0.093 ± 0.05	0.06 ± 0.06	1.41 ± 1.34
	10-RM	0.193 ± 0.06	0.098 ± 0.05	0.07 ± 0.07	1.45 ± 0.63
v_max2_	1-RM	0.285 ± 0.04	0.090 ± 0.05	0.17 ± 0.07	3.02 ± 1.02
	3-RM	0.294 ± 0.04	0.070 ± 0.05	0.19 ± 0.07	2.84 ± 1.08
	6-RM	0.283 ± 0.046	0.086 ± 0.05	0.20 ± 0.07	2.66 ± 1.24
	10-RM	0.294 ± 0.05	0.090 ± 0.06	0.17 ± 0.07	2.43 ± 0.86

## Data Availability

The data presented in this study are available on reasonable request from the corresponding author.

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
