# Peer review of "Comparison of Kinematics and Electromyographic Activity in the Last Repetition during Different Repetition Maximums in the Bench Press Exercise"

_ijerph, 2022, doi:10.3390/ijerph192114238_

Round 1

Reviewer 1 Report

The article analyzes Repetition Maximum for bench press exercises and recommends based on the results which are the most suitable strategy for better performance in competitions that involve similar exercises. The article is interesting. It may be desirable that some technical terms  (p1,F1, etc) are predefined for ease of reading. Some annotations are in the attached document and may be considered while revising the article.

Author Response

The article analyzes Repetition Maximum for bench press exercises and recommends based on the results which are the most suitable strategy for better performance in competitions that involve similar exercises. The article is interesting. It may be desirable that some technical terms  (p1,F1, etc) are predefined for ease of reading. Some annotations are in the attached document and may be considered while revising the article.

Thank you for the comments, we have changed some sentences to get better reading of the article. Furthermore, did we revised annotations like ηp2 and vmin and vmax2.

Reviewer 2 Report

Indicate the SPSS 27 software license

It is a study that meets all the requirements of a scientific process, congratulations.

The limitations are described and leaves the lamp open to continue deepening the knowledge of this area.

It would be interesting to venture with elite lifters to contrast whether the procedure has similar effects than rather inexperienced or low-performance athletes.

Author Response

Indicate the SPSS 27 software license

We have changed this now.

It is a study that meets all the requirements of a scientific process, congratulations.

The limitations are described and leaves the lamp open to continue deepening the knowledge of this area.

Thank you.

It would be interesting to venture with elite lifters to contrast whether the procedure has similar effects than rather inexperienced or low-performance athletes.

We agree that this could be interesting. However, there are more people at the level of the present subjects that we have used than there are elite lifters. Thus, for the major population the findings of the present study are much more interesting than doing the research on elite lifters. But we hope that in future we could do this also in elite lifters.

Reviewer 3 Report

I am grateful for the opportunity to review this manuscript titled "Comparison of kinematics and electromyographic activity in the last repetition during different repetition maximums in the bench press exercise”. The purpose of this study was to investigate the effects of 1-RM, 3-RM, 6-RM, and 10-RM on kinematics and electromyographic activity in the barbell bench press during the last repetition. The data collected in this study may affirm or expand on available literature.

This study is of interest to the IJERPH readers and seems to provide some new findings, applicable to the fields of training. However, the points mentioned in the “Specific comments” section below should be considered and the manuscript amended accordingly before being considered for publication.

Specific comments

1.     I think one of the big problems with the design of this study was that the grip was not standardised for all participants. I would like to hear the authors' views on this.

2.     It would be necessary to include a section on conclusions.

Author Response

I am grateful for the opportunity to review this manuscript titled "Comparison of kinematics and electromyographic activity in the last repetition during different repetition maximums in the bench press exercise”. The purpose of this study was to investigate the effects of 1-RM, 3-RM, 6-RM, and 10-RM on kinematics and electromyographic activity in the barbell bench press during the last repetition. The data collected in this study may affirm or expand on available literature.

This study is of interest to the IJERPH readers and seems to provide some new findings, applicable to the fields of training. However, the points mentioned in the “Specific comments” section below should be considered and the manuscript amended accordingly before being considered for publication.

Specific comments

  1. I think one of the big problems with the design of this study was that the grip was not standardised for all participants. I would like to hear the authors' views on this.

It is a within subject design in which only differences within a subject are investigated. If one subject has a close grip compared to another subject with a wide grip it does not matter, because the grip within a subject was controlled for. Thereby difference between subjects are not interesting. Furthermore, it would be ecological not good if all subjects had the same relative grip, because ths would effect the performance and kinematics of each subject. When controlling for within a subject then evt. differences in kinematics would be the result of lifting load.

  1. It would be necessary to include a section on conclusions.

We have changed this now.